# A Prediction Model for Preoperative Risk Assessment in Endometrial Cancer Utilizing Clinical and Molecular Variables

**DOI:** 10.3390/ijms20051205

**Published:** 2019-03-09

**Authors:** Erin A. Salinas, Marina D. Miller, Andreea M. Newtson, Deepti Sharma, Megan E. McDonald, Matthew E. Keeney, Brian J. Smith, David P. Bender, Michael J. Goodheart, Kristina W. Thiel, Eric J. Devor, Kimberly K. Leslie, Jesus Gonzalez Bosquet

**Affiliations:** 1Compass Oncology, Portland, OR 97227, USA; Erin.Salinas@compassoncology.com; 2Department of Obstetrics and Gynecology, University of Iowa Hospitals and Clinics, Iowa City, IA 52242, USA; marina-miller@uiowa.edu (M.D.M.); kristina-thiel@uiowa.edu (K.W.T.); eric-devor@uiowa.edu (E.J.D.); kimberly-leslie@uiowa.edu (K.K.L.); 3Division of Gynecologic Oncology, Department of Obstetrics and Gynecology, University of Iowa Hospitals and Clinics, Iowa City, IA 52242, USA; andreea-newtson@uiowa.edu (A.M.N.); megan-e-mcdonald@uiowa.edu (M.E.M.); david-bender@uiowa.edu (D.P.B.); michael-goodheart@uiowa.edu (M.J.G.); 4Department of Obstetrics and Gynecology, University of Kentucky, Lexington, KY 52242, USA; dsh274@uky.edu; 5Winfield Pathology Consultants, Central DuPage Hospital, Winfield, IL 60190, USA; mekeeney@llu.edu; 6Department of Biostatistics, University of Iowa College of Public Health, Iowa City, IA 52242, USA; brian-j-smith@uiowa.edu; 7Holden Comprehensive Cancer Center, University of Iowa Hospitals and Clinics, Iowa City, IA 52242, USA

**Keywords:** endometrial cancer, prediction models, high risk, integration of data, clinical outcomes

## Abstract

The utility of comprehensive surgical staging in patients with low risk disease has been questioned. Thus, a reliable means of determining risk would be quite useful. The aim of our study was to create the best performing prediction model to classify endometrioid endometrial cancer (EEC) patients into low or high risk using a combination of molecular and clinical-pathological variables. We then validated these models with publicly available datasets. Analyses between low and high risk EEC were performed using clinical and pathological data, gene and miRNA expression data, gene copy number variation and somatic mutation data. Variables were selected to be included in the prediction model of risk using cross-validation analysis; prediction models were then constructed using these variables. Model performance was assessed by area under the curve (AUC). Prediction models were validated using appropriate datasets in The Cancer Genome Atlas (TCGA) and Gene Expression Omnibus (GEO) databases. A prediction model with only clinical variables performed at 88%. Integrating clinical and molecular data improved prediction performance up to 97%. The best prediction models included clinical, miRNA expression and/or somatic mutation data, and stratified pre-operative risk in EEC patients. Integrating molecular and clinical data improved the performance of prediction models to over 95%, resulting in potentially useful clinical tests.

## 1. Introduction

Endometrial cancer is the most common gynecologic cancer diagnosed in the United States, with an estimated 61,380 new cases and 10,920 deaths in 2017 [1]. Endometrioid endometrial cancer (EEC) is the most common histologic subtype. More than half of the patients with EEC are considered to be low risk, diagnosed at an early stage [2], and will not benefit from comprehensive surgical staging [3,4] or require further treatment after surgery [5]. However, there is a subset of patients at a higher risk of having extrauterine disease, an increased risk of recurrence, need for adjuvant treatment, and poorer overall prognosis. Currently, there is no accurate way to correctly identify these high risk patients pre-operatively, and the prognosis and survival of patients is based on information obtained during surgery [6].

Prediction models based on lymph node (LN) involvement to identify high risk endometrial cancer patients have a positive predictive value of around 20% in EEC [7,8] and rely on uterine factors obtained intra-operatively and/or on frozen pathologic evaluation. Using these algorithms for prediction of LN involvement, it would be necessary to perform 4 to 8 lymphadenectomies to find one patient with true positive LNs [9]. There are major risks associated with surgical staging and LN dissection; these include increased operative time, potential for blood loss associated with vascular injury, genitofemoral nerve injury, lymphocyst formation, and lymphedema [10,11,12,13]. In an effort to decrease the morbidity associated with full LN dissections, sentinel lymph node (SLN) biopsy was evaluated to predict risk. Mapping identified at least one SLN in 81%–86% of patients who had disease in 10%–12% of LNs [14,15]. Thus, the SLN biopsy strategy is aimed to detect LN status as a proxy for higher EEC risk, though variables associated with the level of risk are available only after surgical treatment and full pathologic evaluation. In performing SLN biopsies, we are exposing low risk patients to additional surgery who could be cured with hysterectomy alone. A benefit of the SLN biopsy technique is seen in high risk endometrial cancer, or high risk types like carcinosarcoma, serous, and clear cell carcinomas, where prospective studies have shown that SLN biopsy is a reasonable alternative to complete LN dissection [16]. LN involvement is only one of the independent risk factors for poorer outcomes in EEC. For example, involvement of the adnexa, lymphovascular involvement, involvement of the cervix and other distant organs are also independent risk factors for disease recurrence and death, even in the absence of LN invasion; these features can only be ascertained after the surgical specimens are processed. Moreover, recurrence occurs in up to 8% of EEC patients with none of these risk factors [10].

Clinical and pathologic prognostic factors that could predict outcomes in EEC have been validated both retrospectively and prospectively [6,17,18]. Prediction models constructed using clinical prognostic factors, like age and histologic grade, have a good performance, with an area under the curve (AUC) ranging from 75%–82% [19]. With the advent of rapid sequencing of tumors and publicly available genomic datasets like The Cancer Genome Atlas (TCGA), molecular-based prediction models are now possible. Molecular heterogeneity is thought to underlie the observed differences in clinical phenotypes. Based on this concept, patterns of gene expression have proven useful in the prediction of clinical phenotypes in several cancers, including breast and ovarian [20,21,22]. We hypothesized that by integrating thorough clinical data and molecular characteristics from biopsy specimens, we will be able to create prediction models in EEC that can be used to stratify patients into risk groups prior to surgery. This would in turn guide preoperative counseling and surgical management to determine which patients would benefit from less surgery (hysterectomy alone) versus more surgery with LN evaluation and staging.

The aim of the study herein was to create the best performing prediction model to classify EEC patients by risk using a combination of molecular and clinical-pathological variables available from our institution, the University of Iowa (UI), and from publicly available genomics data repositories (TCGA and Gene Expression Omnibus (GEO)). With this objective in mind, we constructed risk prediction models integrating multiple classes of molecular data with clinical-pathologic information that is available prior to surgery.

## 2. Results

The flowchart of patients included in the UI analysis is represented in Figure 1. Clinical and pathological characteristics of these patients are described in Table 1.

### 2.1. Survival Analysis

Five-year survival was 98%for UI low risk EEC patients and 75% for high risk patients (*p* < 10^−4^, Figure 2A). For TCGA dataset, five-year survival was 95% for low risk patients and 75% for high risk EEC patients (Figure 3, *p* < 10^−4^). In a multivariate analysis of the UI dataset, previous stroke, number of positive LN, and risk level were independently associated with disease-specific survival (*p* < 0.05, Figure 2B). These analyses demonstrate that level of risk is an independent outcome measure and that low risk patients have excellent five-year survival.

### 2.2. Variable Selection for Prediction Modeling

RNA extracted from primary tumor tissue samples of 62 EEC patients was of sufficient quality for subsequent RNA sequencing (RNA-seq, Figure 1). This sub-cohort included tissue from 26 high risk patients and 36 low risk patients. RNA-seq analysis resulted in gene expression data for 26,336 genes and 1916 micro-RNAs (miRNAs), along with identification of 12,340 somatic mutations and 26,720 segments with gene copy number variations (CNV). Cross-validation selection analysis identified 255 genes, 55 miRNAs, 398 somatic mutations and 846 CNVs that were most informative in the prediction process (Figure 4). Only those variables selected by cross-validation were included in prediction analyses.

### 2.3. Prediction Models

#### 2.3.1. UI Prediction Models

To build models that include only one type of data, or data class, we used only the variables that were selected with the cross-validation analysis. Accordingly, the input variables for models with one type of data were built using 17 clinical variables, 255 mRNAs, 55 miRNAs, 398 somatic mutations and 846 CNVs. The prediction analysis with Lasso discarded those variables that had no influence in the prediction model and incorporated the most informative, or resulting variables, for the prediction process: 7 clinical variables, 38 mRNAs, 28 miRNAs, 35 somatic mutations, and 65 CNVs (Table 2, Prediction models including one data class). For prediction analyses using more than one data class, we used the resulting variables and integrated them in the same model, as these were the best predictors for that data class (Table 2, Prediction models including two, three, four and five data classes). Each prediction model with more than one data class had different combinations of resulting variables, depending on which were more informative for that particular integrative model (Table 2).

A prediction model only including clinical variables had a performance of 88%, as measured by the AUC (model M1-A, Table 2). Models of selected molecular variables (gene expression, miRNA expression, CNVs or somatic mutations) did not perform as well. However, integrating clinical data with one or more of the molecular data categories improved prediction performance by 10–15% as compared to models with only single classes of selected molecular variables (Table 2). The best prediction models included clinical data combined with miRNA expression and/or somatic mutations (models M2-B, M2-C and M3-C in Table 2, Figure 5). Although prediction models including selected variables from gene expression performed fairly in UI models, we were not able to replicate those results in the validation set.

#### 2.3.2. TCGA/GEO Replication

To assess how UI models perform in independent datasets, we replicated the analysis with the same variables that were selected by cross-validation in the UI database and used those variables in TCGA and GEO datasets (Table 3). While prediction models including only selected clinical variables performed rather well, other models with only molecular variables, as well as models with combinations of clinical and molecular variables, performed 10 to 20 AUC percentage points lower than UI models. This could be explained in part because some of the variables in the UI dataset were not available in TCGA and GEO datasets. Another factor that must be considered is that patients in the different cohorts are abstracted from different populations that may have dissimilar genetic background compositions. Better performing models included clinical data and selected CNVs, with somatic mutations and/or miRNA variables, all had AUC performances of over 75% (TCGA model M3-B, TCGA model M3-E, and TCGA model M4-D in Table 3). Models including gene expression did not replicate as well. Part of this decline in performance may be because two selected transcripts in the UI model (*FAM134B* and *LOC101927701*, a non-coding RNA) had no expression data in TCGA dataset.

#### 2.3.3. TCGA Validation

For external validation of UI prediction models in TCGA data, we chose models with high performance using the UI dataset that also replicated the best in TCGA dataset: (1) Model M2-B: clinical and miRNA data; (2) model M2-C: clinical and somatic mutation data; (3) model M3-C: clinical, gene expression (mRNA), and somatic mutation data; and (4) model M3-D: clinical, miRNA, and somatic mutation data (Table 3 and Table 4). For each model, we defined a threshold, or cut-off (Table 4), which is necessary to test the predictive accuracy of the models. Thresholds are set at the values for the score of the model above which patients were classified as high risk with a sensitivity of >90% (see also Section 4). Models that were best validated in an independent database (TCGA) included clinical and miRNA data (M2-B) and somatic mutations (M3-D), with accuracies of around 60%, and negative predictive values (NPV) of >75%. Table 5 details which variables were used in validated models. Scores for each model are calculated by multiplying the value of the clinical variable, normalized miRNA expression, normalized gene expression, or number of somatic mutations by the weight of the variable, followed by adding each of these weighted values (for additional details, see Table A1, Table A2 and Table A3).

## 3. Discussion

Our study aimed to create a prediction model that could have an immediate impact on treatment decisions. As anticipated, five-year survival was significantly associated with our definitions of high risk (HR) and low risk (LR) patients in both our UI patient database and in TCGA dataset for EEC. In our prior study using somatic mutations and variant allele frequencies, we created a prediction model with an AUC of 91% [19]. The limitation of that model was the technology required to determine allele frequencies of mutated genes. Selected genes had to be sequenced to a depth of at least 50× to have an accurate allele frequency determination, a process requiring considerable time and expense. Herein, we built on this concept of integrating detailed clinical and molecular data by expanding the classes of molecular data that were used in the models. With the addition of more variables associated with levels of risk, we substantially improved our model performance based on AUC values.

Although clinical data alone had a performance of 88% to predict if a patient will be high risk, the best prediction models combined clinical and miRNA expression data, with or without somatic mutations. Based on our results, simply adding more data classes did not substantially improve the models. Rather, finding the optimal molecular data to include in combination with comprehensive and accurate clinical data creates the best prediction model. The addition of molecular markers is an improvement over our current pre-operative models that use only clinical-pathological markers, grade and age (95% CIs are not overlapping), which solidifies our contention that molecular parameters are necessary to improve our clinical ability to predict which patients are high risk prior to surgery.

Validation in TCGA and GEO databases indicated consistency in model performance. However, performance in the validation models was overall lower, which may be attributed to less comprehensive clinical data in those databases. For example, comparing only clinical data resulted in a performance of 75% using TCGA clinical-pathological features versus 88% using the UI dataset. This supports the importance of obtaining thorough clinical information. In addition, patients in TCGA, GEO and UI datasets were recruited from different populations that may have different genetic backgrounds. For example, in a study aimed to determine the genetic substructure of the population being recruited, we observed that in our institution we only identified one subpopulation, mostly of European origin [23]. However, 4–6 subpopulations with a more diverse background were identified in patients included in the TCGA dataset. Patient population sub-structure can therefore limit the utility of a database to validate results from a single institution. However, consistently higher performances in the training set (UI) than in the testing sets (TCGA, GEO) could also be due to overfitting, which is a common problem that occurs when there are more variables in the analysis than in the samples [24], despite using adequate methods and resampling (cross-validation techniques). Overfitting may contribute to overoptimistic results in some prediction models, and could also be minimized in future prospective validations.

A strength of our prediction model is that it can distinguish between low risk patients and high risk patients that would likely benefit from a more aggressive surgical approach, surgical staging, and adjuvant treatment. Our prediction models for high risk patients are not based only on LN involvement. We also included as high risk patients those with adnexal or cervical involvement, or patients with other local and/or distant metastases. These patients have poorer prognosis, despite a lack of LN involvement [10]. The resulting models have the potential to be more comprehensive with more coverage than those only including LN involvement as a high risk factor, like those models based on SLN biopsies [15,16]. This is especially important for obese patients with multiple medical comorbidities, who are at increased surgical risk. A pre-operative risk stratification test would be highly useful to guide management and tailor pre-operative discussions, risk evaluation, and surgical planning.

Our study was performed on tumors from hysterectomy specimens, with access to all dissected tumor material. We acknowledge that there are some feasibility issues that must be addressed before these prediction models could be implemented as a laboratory test to prospectively predict risk using biopsy specimens. For example, uterine biopsies may not properly represent the heterogeneity of the whole tumor, which may influence interpretation of tumor grade. However, retrospective studies have demonstrated that molecular parameters are highly concordant between diagnostic preoperative biopsies and surgical specimens [25]. Molecular-based prediction models from biopsy specimens will likely be consistent with prediction models created with surgical specimens. The relatively low amount of material from biopsy specimens is not a concern because sequencing can be accomplished using small RNA quantities, and with rapid turn-around time that would allow for risk prediction prior to surgery (typically 3–4 weeks after biopsy) [25,26]. We envision that the ideal test would utilize fresh tissue from the uterine biopsy and polymerase chain reaction (PCR) to gather molecular data. Such a test would therefore be quick, affordable, and widely available. Finally, our validation strategy used retrospective data, and prospective validation of these initial prediction models is necessary. An additional strength of our study is that it was validated using TCGA, one of the most comprehensive, publicly available databases from EEC specimens.

Variables in the best prediction models can be considered classifiers for high risk patients. A classifier can be used to select and stratify patients for therapy. However, the components of a classifier are not necessarily biological markers of disease severity [27]. Indeed, there have been other attempts to stratify EEC based on tumor characteristics. Tumors in the TCGA study were grouped based on similar features by clustering, which is an unsupervised learning method [28]. Other histological types of endometrial cancer, such as the more aggressive serous type, were used to build these clusters, but poor prognosis or clinical outcomes of these patients were not included. Moreover, clustering is different than classification. Classification is a supervised learning method that assigns previously predefined labels based on molecular features. One suggestion is to perform classification of risk based on TCGA endometrial molecular groups built with clustering: POLE ultramutated, microsatellite instability hypermutated, copy-number low and copy-number high [28]. Accuracy of those TCGA-based models range from 59% to 73% [29,30,31]. Models described herein better predict high risk status in EEC patients, with AUC >90%.

Finally, an additional benefit of our integrated models is the identification of putative mechanisms of risk, which are suggested by the specific molecular variables in the best models. For example, miRNAs such as MIR-181, MIR-30b, and MIR-200b as well as specific gene loci such as *ADAMTS13*, *NOTCH4* and *PIGN* have been linked to many cancers, including EEC [32], and should be evaluated in vitro in the future.

## 4. Materials and Methods

### 4.1. Classification of EEC Risk

Classification of EEC risk was based on the criteria and results from Gynecologic Oncology Group (GOG) 33 and GOG 99 clinical trials [6,17]. High risk patients were defined as those presenting with stage II, III and IV disease as defined by 2009 FIGO classification and sanctioned in 2014 [33] as well as patients with stage I disease and high-intermediate risk features as defined by GOG 99 criteria [17]. High intermediate risk features in stage I tumors included grade 2 or 3 tumors, presence of lymphovascular invasion, and outer-third myometrial invasion with the following criteria: (1) At least 70 years of age with at least one risk factor; (2) at least 50 years of age with two risk factors; and (3) any age with all three risk factors. Low risk patients include all other stage I patients either with no myometrial invasion or low-intermediate risk features as defined by GOG 99 criteria [17]. There were 206 low risk and 194 high risk patients from TCGA dataset and 70 low risk and 56 high risk patients in the UI dataset.

### 4.2. Patients and Clinical Data Collection

#### 4.2.1. University of Iowa (UI)

Endometrial cancer patients with endometrioid histology and complete clinical and pathological data were included. Patients with secondary gynecologic malignancies, neoadjuvant chemotherapy or radiation, and/or incomplete data were excluded. A total of 70 low risk patients and 56 high risk patients were identified and included in the validation analysis. One patient had no information about level of risk recorded and was excluded from the analysis. An outline of the study population is shown in Figure 1. Clinical and pathological characteristics are described in Table 1.

The institutional review board (IRB) of the UI approved the current study including human subjects/materials on 28 July 2016 (IRB Number 201607815: ‘*Prediction Model for Risk Assessment in Endometrial Cancer*’).

#### 4.2.2. The Cancer Genome Atlas (TCGA)

Patients with non-endometrioid histology were excluded. Of those patients with Type I endometrial cancer, or EEC, clinical and RNA-seq data were downloaded. Patients were divided into risk categories, high risk (N = 194) and low risk (N = 206), as described above. Clinical and pathological characteristics of TCGA EEC patients included in the analysis are provided in Appendix C
Table A4. The distribution between low and high risk patients in UI and TCGA cohorts was similar (chi square *p*-value = 0.33).

#### 4.2.3. Gene Expression Omnibus (GEO)

Only clinical data and gene expression data from a microarray expression experiment were available from this dataset, reference number GSE17025 [34] (Table A5).

### 4.3. Biological Data

#### University of Iowa (UI)

*RNA Purification and Sequencing:* The University of Iowa Department of Obstetrics and Gynecology maintains a Women’s Health Tissue Repository (WHTR) containing more than 60,000 biological samples, including more than 2500 primary gynecologic tumors [35]. All tissues in the WHTR are collected under informed consent (IRB#200910784 and IRB#200209010). Of the 126 patients identified in the original EEC panel, we were able to obtain 62 primary tumor tissues with sufficient RNA yield and quality for analysis, 36 low risk and 26 high risk (no differences in distribution relative to the complete patients’ cohort: chi square *p*-value = 0.74).

Total cellular RNA was purified from primary tumor tissue using the mirVana (Thermo Fisher, Waltham, MA, USA) RNA purification kit following manufacturers’ instructions. Yield and quality of purified cellular RNA was assessed using a Trinean DropSense 16 spectrophotometer and an Agilent Model 2100 bioanalyzer. Only RNAs with an RNA integrity number (RIN) [36] greater than or equal to 7.0 were selected for RNA sequencing.

Equal mass total RNA (500 ng) from each qualifying tumor was fragmented, converted to cDNA and ligated to bar-coded sequencing adaptors using Illumina TriSeq stranded total RNA library preparation (Illiumina, San Diego, CA, USA). Molar concentrations of the indexed libraries were confirmed on the Agilent Model 2100 bioanalyzer and libraries were then combined into equi-molar pools for sequencing. The concentration of the pools was confirmed using the Illumina Library Quantification Kit (KAPA Biosystems, Wilmington, MA, USA). Sequencing was then carried out on the Illumina HiSeq 4000 genome sequencing platform using 150bp paired-end SBS chemistry. All library preparation and sequencing was performed in the Genome Facility of the University of Iowa Institute of Human Genetics (IIHG).

File pre-processing of diverse biological data: Briefly, sequence reads were mapped and aligned to the human reference genome (version hg38) using STAR, a paired-end enabled algorithm [37]. BAM files were produced after alignment. We used featureCount to measure gene expression from BAM files [38]. After the gene counts were generated, we used DESeq2 package to import, normalize and prepare the data for analysis [39]. We independently used gene expression and miRNA expression for the association analysis.

BAM files for each sample were also used for mutation discovery and base-calling against the human genome reference utilizing SAMtools and BCFtools [40]. Results were annotated with ANNOVAR and formatted to display the number of mutations per gene and sample [40]. We included only non-synonymous somatic mutations. CNV was determined with SAMtools and CopywriteR using BAM files as input [41].

### 4.4. Statistical Analysis

#### 4.4.1. Survival Analysis

To assess the association of survival with risk levels and other clinical variables, survival analysis was performed using Cox proportional hazard ratios. All variables associated with survival in a univariate analysis (*p* ≤ 0.05) were included in the multivariate regression model.

#### 4.4.2. Variable Selection for Prediction Modeling

In the prediction model, we only used those variables that could be assessed at baseline, prior to initiation of treatment. Our approach was to (1) reduce the number of variables using a univariate selection of prediction variables with cross-validation; (2) introduce those significant variables from the univariate selection process to the prediction model of level of risk. Rather than introducing all variables directly in the prediction model, this approach was chosen because it would likely lead to a model that is more sparse (i.e., simpler, with fewer variables) and can be more easily validated retrospectively and prospectively. To reduce the number of variables, we used the *caret* R package [42]. This software fits a simple linear model between a single feature and the outcome. Features that were statistically significant (*p*-values < 0.05) in this univariate analysis were then used for multivariate Lasso regression modeling. Cross-validation of the subsequent models will be biased unless some sort of resampling is included in the feature selection step [43]. Thus, variable selection for all classes of clinical and biological data (gene and miRNA expression, gene copy number and mutation analysis) were performed using k-fold cross-validation with the *caret* R package to decrease the possibility of overfitting the final model [42].

#### 4.4.3. Prediction Model Construction

Selected clinical and molecular variables from the k-fold cross-validation process were analyzed individually and in combination to determine their prediction potential for preoperative risk. The Lasso method, as implemented in the glmnet R package [44], was used to develop a regression model to predict low risk versus high risk patients. We selected Lasso because it is a multivariate regression method that allows simultaneous selection and estimation of the effects of variables, while accounting and adjusting for confounding factors. In our experience, Lasso consistently handles missing values and lower number of samples and computes the AUC without reporting any errors, as compared to other prediction methods [45]. We evaluated the performance of our model using the AUC and its 95% CI. AUC was estimated with 1000 replicates of 10-fold cross-validation to avoid over-fitting of the model (internal validation) [27]. Bias-corrected and accelerated bootstrap CIs were computed for resulting AUCs. A value of 0.5 indicates a lack of model predictive performance, and 1.0 indicates perfect predictive performance.

#### 4.4.4. The Cancer Genome Atlas Replication and Validation

We replicated (or repeated) the same analysis performed in the UI dataset in TCGA dataset. For this external replication, we included the same variables used in modeling with the UI dataset, but extracted data from TCGA datasets. Then, the same Lasso analysis used in the UI cohort was performed in this TCGA data. The performance of the replication analysis was measured in terms of AUC and 95% CIs.

For validation of UI prediction models in the TCGA dataset, we took the models built using UI dataset and inserted TCGA data to see if they could discriminate between low and high risk patients in the TCGA dataset. The validation analysis does an inference: with the UI-built model and TCGA data, the model attempts to predict low or high risk classes. For the validation analysis, we took the best UI prediction models for risk that replicated well in TCGA and applied them to the TCGA dataset to obtain a predicted probability of high risk for each patient [44]. Then, we used the R package *pROC* to determine thresholds, or cut-offs, for the UI model applied to the TCGA data [46]. Thresholds were treated as a tuning parameter for which values were sought to produce a final classification model and were computed with 2000 bootstrap replicates. Threshold values that yielded sensitivities around 90% were ranked from highest to lowest sensitivity and negative predictive value. Among the ranked results, the top-ranked set of tuning parameters was used to fit a final score of the model to the entire set of patients and define the classification rule. The goal was to create models that would identify at least 90% of high risk patients, while ruling out most patients with low risk.

An example of R coding for the variable selection, Lasso prediction analysis, replication and validation analysis is provided in a Appendix A.

## 5. Conclusions

Combining clinical and molecular data on EEC tumor specimens allows us to stratify patients into high and low risk categories with greater than 95% confidence. The performance of our prediction model is superior to the current standard of care using clinical factors alone. Identification of crucial molecular variables from next generation technologies can be developed using conventional and quantitative PCR and expression arrays to enable design of a novel diagnostic test to be used on tissue obtained from endometrial biopsies.

## Figures and Tables

**Figure 1 ijms-20-01205-f001:**
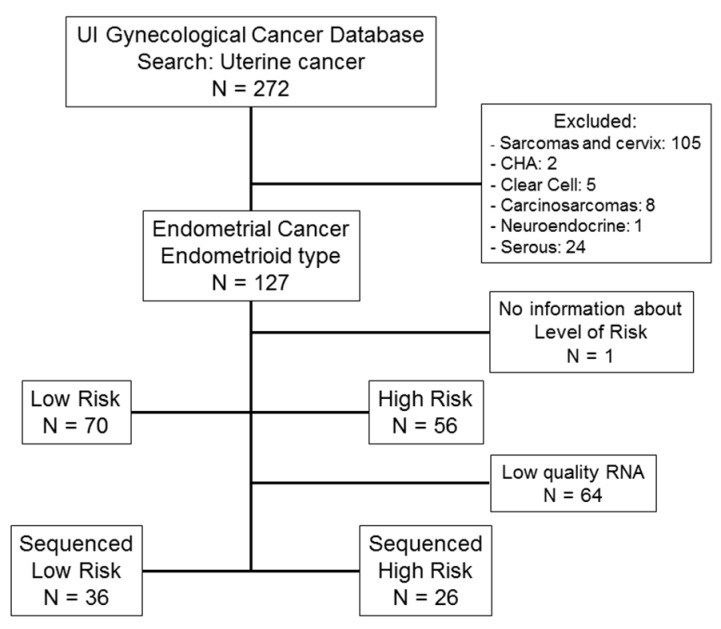
Flow chart of patients included in the University of Iowa (UI) endometrial cancer study cohort. (CHA = complex endometrial hyperplasia with atypia). In this dataset, 126 patients had endometrial cancer, the endometrioid type. Only 62 had sufficient quantity and quality of purified RNA for RNA sequencing.

**Figure 2 ijms-20-01205-f002:**
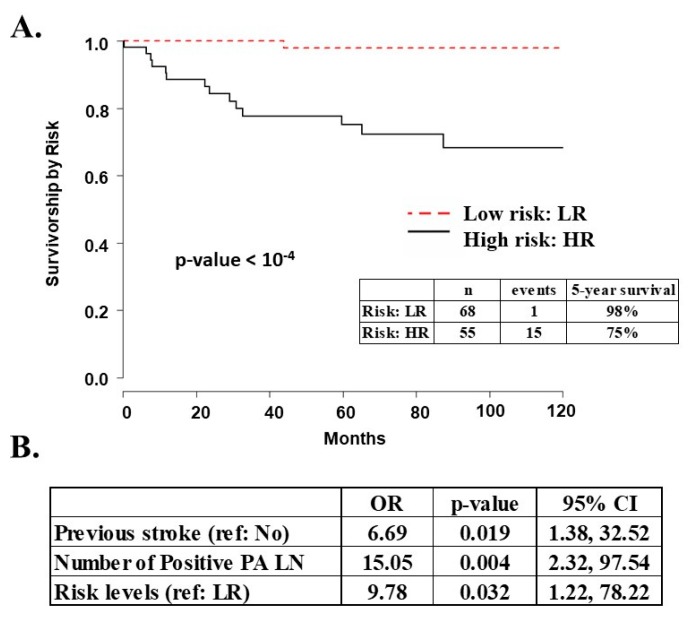
Survival analysis for UI EEC patients. (**A**) Survival curves for UI EEC patients with clinical data stratified by risk. There were two low risk and one high risk patients with no survival information; (**B**) Independent variables associated with survival in the multivariate analysis for UI EEC patients. Ref: reference value; PA LN: Para-aortic lymph nodes; LR: low risk. High risk patients have almost 10 times greater risk of dying from endometrial cancer relative to low risk patients.

**Figure 3 ijms-20-01205-f003:**
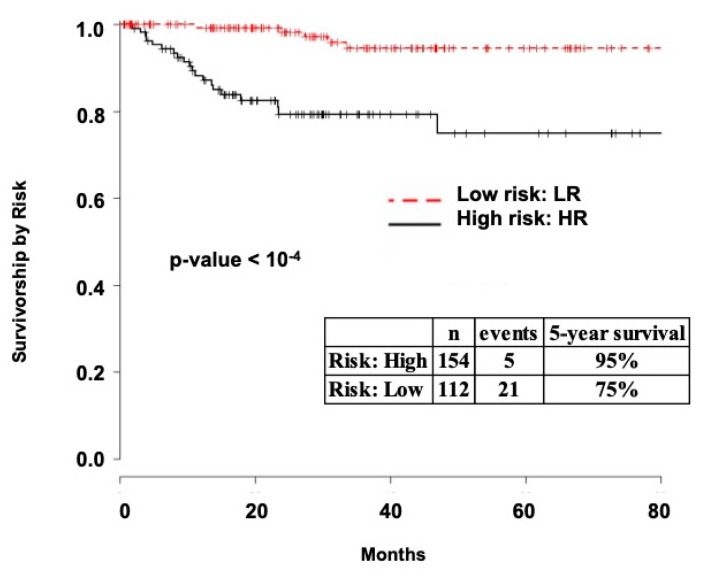
Survival analysis for TCGA endometrial cancer patients. Survival curves for TCGA EEC patients with clinical data stratified by risk. Survival for both UI and TCGA patients was similar when stratified by risk level.

**Figure 4 ijms-20-01205-f004:**
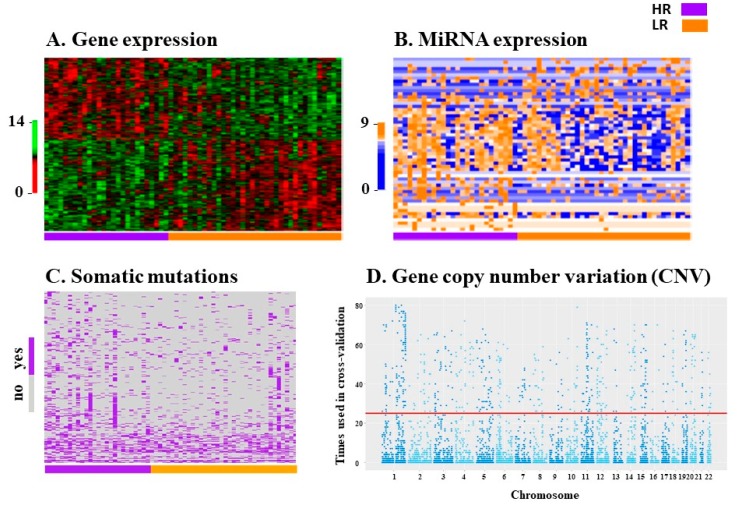
Selection of molecular variables for prediction model analyses for both groups, low risk (LR (N = 36), in orange) and high risk (HR (N = 26), in purple). Variables that passed a cut-off *p*-value < 0.05 in a univariate linear model and were present in each fold of the k-fold cross-validation were selected. In **A**, **B** and **C**: patients are on the *X* axis and molecular variables are on the *Y* axis. (**A**) Heatmap of expression of 255 selected genes out of a total of 26,336 genes. Normalized gene expression is represented in a red-green scheme from lower to higher expression, respectively; (**B**) Heatmap of 55 selected miRNAs out of a total of 1916 miRNAs. Normalized miRNA expression is represented in a blue-orange scheme from lower to higher expression, respectively; (**C**) Heatmap of 398 selected somatic mutations out of a total of 12,340 mutations. Each somatic mutation for each patient is represented in purple. Grey represents non-mutated genes; (**D**) Manhattan plot of 846 selected loci with copy number variation out of a total of 26,720 loci. The *Y* axis represents how many times the locus was involved in the prediction process with cross-validation (k-fold with 25 replications). The *X* axis represents the chromosomal location. The horizontal red line denotes 25 replications. See the Variable Selection Section 4 for more details.

**Figure 5 ijms-20-01205-f005:**
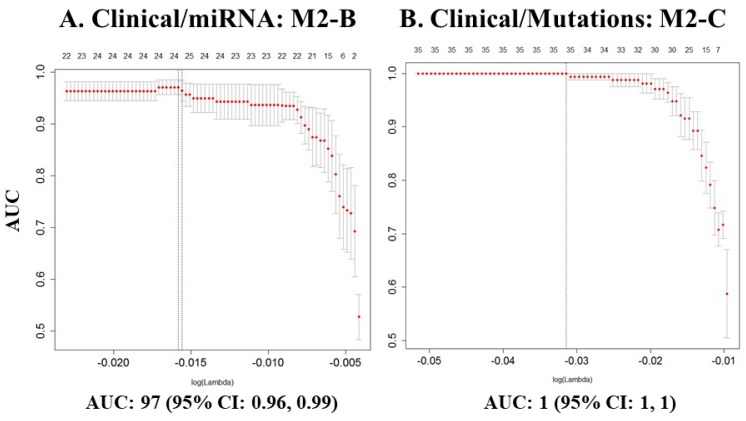
Prediction models with the highest performances. Curves of performance of the models based on AUC. On top of the graphic: number of variables included in the model. On *Y*-axis: AUC value. On *X*-axis: lambda value. (**A**) Model including clinical and miRNA variables; (**B**) Model including clinical variables and somatic mutations.

**Table 1 ijms-20-01205-t001:** Patient clinical and pathological characteristics. Univariate analysis with logistic regression was used to assess differences between both groups. * denotes statistically significant differences between low and high risk patients.

	Clinical/Pathological Variables	Low Risk (N = 70)	High Risk (N = 56)	*p*-Value
Preoperative characteristics	Age (mean)	58.7	64.8	0.003 *
	BMI (mean)	38.5	32.6	<0.001 *
	Charlson Morbidity Index (mean)	4.7	5	0.012 *
	Grade			<0.001 *
	1	38	7	
	2	21	27	
	3	8	22	
Postoperative characteristics	Invasion (mean)	19	62	<0.001 *
	2009 FIGO Stage			0.991
	I	70	23	
	II	-	7	
	III	-	20	
	IV	-	6	
	Lymph nodes (% positive)	0 (0%)	13 (27%)	0.987
	Peritoneal Cytology (% positive)	2 (3%)	31 (56%)	0.011 *
	Lymphovascular involvement (% positive)	2 (3%)	10 (19%)	<0.001 *
	ER (% positive)	38 (93%)	31 (78%)	0.066
	PR (% positive)	38 (93%)	30 (75%)	0.040 *
	Postoperative complications (% positive)	12 (17%)	17 (32%)	0.056
	LOS (mean days)	3.3	6.1	0.002 *
	Adjuvant Treatment (yes) (% positive)	8 (11%)	39 (74%)	<0.001 *
Outcomes	5-year Survival (%)	98%	75%	<0.001 *
	Recurrence (% positive)	2 (3%)	19 (37%)	<0.001 *
	Death due to disease (% positive)	1 (1%)	15 (30%)	0.001 *

**Table 2 ijms-20-01205-t002:** Prediction models for levels of risk using diverse clinical, pathological and molecular data. Models that included only 1 data class used all variables selected with the cross-validation analysis (input variables: 17 clinical features, 255 mRNAs, 55 miRNAs, 398 somatic mutations and 846 CNVs). The Lasso analysis selected only the most informative resulting variables for the prediction process: 7 clinical features, 38 mRNAs, 28 miRNAs, 35 somatic mutations, and 65 CNVs. For prediction analysis using more than one data class, we used the resulting variables. Model performances were measured by AUC and their 95% confidence interval (CI). Models with the best performance are marked with * Prediction models using only one data class. # Number of variables

**Model Number**	**Data Class**	**# Input Variables**	**# Resulting Variables**	**AUC**	**95% CI**
M1-A	Clinical	17	7	0.88	0.84, 0.92
M1-B	mRNAs	255	38	0.79	0.73, 0.85
M1-C	miRNAs	55	28	0.84	0.76, 0.93
M1-D	Mutations	398	35	0.68	0.63, 0.73
M1-E	CNVs	846	65	0.67	0.56, 0.77
**Prediction Models Using Two Data Classes**					
**Model Number**	**Data Classes Included** **Clinical +**	**# Input Variables**	**# Resulting Variables**	**AUC**	**95% CI**
M2-A	mRNAs	7 + 38	37	0.93	0.90, 0.96
* M2-B	miRNAs	7 + 28	24	0.97	0.96, 0.99
* M2-C	Mutations	7 + 35	35	1	1, 1
M2-D	CNVs	7 + 65	61	0.92	0.89, 0.94
**Prediction Models Using Three Data Classes**					
**Model Number**	**Data Classes Included** **Clinical +**	**# Input Variables**	**# Resulting Variables**	**AUC**	**95% CI**
M3-A	mRNAs + miRNAs	7 + 38 + 28	37	0.83	0.74, 0.91
M3-B	Mutations + CNVs	7 + 35 + 65	48	0.94	0.91, 0.97
M3-C	mRNAs + Mutations	7 + 38 + 35	41	0.95	0.92, 0.98
M3-D	miRNAs + Mutations	7 + 28 + 35	36	0.94	0.91, 0.97
M3-E	miRNAs + CNVs	7 + 28 + 65	46	0.86	0.81, 0.91
M3-F	mRNAs + CNVs	7 + 38 + 65	44	0.93	0.91, 0.95
**Prediction Models Using Four Data Classes**					
**Model Number**	**Data classes Included** **Clinical +**	**# Input Variables**	**# Resulting Variables**	**AUC**	**95% CI**
M4-A	mRNAs + miRNAs + Mutations	7 + 38 + 28 + 35	42	0.94	0.91, 0.96
M4-B	mRNAs + miRNAs + CNVs	7 + 38 + 28 + 65	40	0.91	0.88, 0.93
M4-C	mRNAs + Mutations + CNVs	7 + 38 + 35 + 65	42	0.91	0.88, 0.95
M4-D	miRNAs + Mutations + CNVs	7 + 28 + 35 + 65	53	0.88	0.84, 0.92
**Prediction Models Using Five Data Classes**					
**Model Number**	**Data Classes Included Clinical +**	**# Input Variables**	**# Resulting Variables**	**AUC**	**95% CI**
M5-A	mRNAs + miRNAs + Mutations + CNVs	7 + 38 + 28 + 35 + 65	47	0.89	0.86, 0.92

**Table 3 ijms-20-01205-t003:** External replication of prediction models for levels of risk. In the analysis of TCGA and GEO datasets, we used resulting variables from UI analyses of 1 data class or type; these results are included for comparison in this table and denoted as “UI model #”) (The definitions for input variables and resulting variables are the same as in Table 2). In most cases, variables resulting from the UI analyses were not available in external sets (marked by *). Model performances were measured by AUC and their 95% confidence interval (CI).

**Replication of Prediction Models Using One Data Class**					
**Model Number**	**Data Class**	**# Input Variables**	**# Resulting Variables**	**AUC**	**95% CI**
**UI model M1-A**	**Clinical**	**17**	**7**	**0.88**	**0.84, 0.92**
TCGA model M1-A	Clinical	2 *	2	0.75	0.73, 0.78
GEO model M1-A	Clinical	2 *	2	0.84	0.79, 0.89
**UI model M1-B**	**mRNAs**	**255**	**38**	**0.79**	**0.73, 0.85**
TCGA model M1-B	mRNAs	36 *	23	0.60	0.57, 0.63
GEO model M1-B	mRNAs	14 *	5	0.60	0.53, 0.68
**UI model M1-C**	**miRNAs**	**55**	**28**	**0.84**	**0.76, 0.93**
TCGA model M1-C	miRNAs	28	4	0.57	0.54, 0.60
**UI model M1-D**	**Mutations**	**398**	**35**	**0.68**	**0.63, 0.73**
TCGA model M1-C	Mutations	34 *	18	0.59	0.57, 0.62
**UI model M1-C**	**CNVs**	**846**	**65**	**0.67**	**0.56, 0.77**
TCGA model M1-E	CNVs	65	2	0.63	0.59, 0.67
**Replication of Prediction Models Using Two Data Classes**					
**Model Number**	**Data Classes Included** **Clinical +**	**# Input Variables**	**# Resulting Variables**	**AUC**	**95% CI**
**UI model M2-A**	**mRNAs**	**7 + 38**	**37**	**0.93**	**0.90, 0.96**
TCGA model M2-A	mRNAs	2 + 36 *	15	0.75	0.72, 0.78
GEO model M2-A	mRNAs	2 + 14 *	2	0.92	0.90, 0.95
**UI model M2-B**	**miRNAs**	**7 + 28**	**24**	**0.97**	**0.96, 0.99**
TCGA model M2-B	miRNAs	2 + 28 *	3	0.75	0.72, 0.77
**UI model M2-C**	**Mutations**	**7 + 35**	**35**	**1**	**1, 1**
TCGA model M2-C	Mutations	2 + 34 *	30	0.75	0.73, 0.77
**UI model M2-D**	**CNVs**	**7 + 65**	**61**	**0.92**	**0.89, 0.94**
TCGA model M2-D	CNVs	2 + 65 *	3	0.75	0.71, 0.79
**Replication of Prediction Models Using Three Data Classes**					
**Model Number**	**Data Classes Included** **Clinical +**	**# Input Variables**	**# Resulting Variables**	**AUC**	**95% CI**
**UI model M3-A**	**mRNAs + miRNAs**	**7 + 38 + 28**	**37**	**0.83**	**0.74, 0.91**
TCGA model M3-A	mRNAs + miRNAs	2 + 36 + 28 *	4	0.75	0.72, 0.78
**UI model M3-B**	**Mutations + CNVs**	**7 + 35 + 65**	**48**	**0.94**	**0.91, 0.97**
TCGA model M3-B	Mutations + CNVs	2 + 34 + 65 *	24	0.78	0.75, 0.80
**UI model M3-C**	**mRNAs + Mutations**	**7 + 38 + 35**	**41**	**0.95**	**0.92, 0.98**
TCGA model M3-C	mRNAs + Mutations	2 + 36 + 34 *	2	0.74	0.71, 0.77
**UI model M3-D**	**miRNAs + Mutations**	**7 + 28 + 35**	**36**	**0.94**	**0.91, 0.97**
TCGA model M3-D	miRNAs + Mutations	2 + 28 + 34 *	2	0.74	0.72, 0.75
**UI model M3-E**	**miRNAs + CNVs**	**7 + 28 + 65**	**46**	**0.86**	**0.81, 0.91**
TCGA model M3-E	miRNAs + CNVs	2 + 28 + 65 *	5	0.76	0.73, 0.79
**UI model M3-F**	**mRNAs + CNVs**	**7 + 38 + 65**	**44**	**0.93**	**0.91, 0.95**
TCGA model M3-F	mRNAs + CNVs	2 + 36 + 65 *	2	0.75	0.72, 0.78
**Replication of Prediction Models Using Four Data Classes**					
**Model Number**	**Data Classes Included** **Clinical +**	**# Input Variables**	**# Resulting Variables**	**AUC**	**95% CI**
**UI model M4-A**	**mRNAs + miRNAs + Mutations**	**7 + 38 + 28 + 35**	**42**	**0.94**	**0.91, 0.96**
TCGA model M4-A	mRNAs + miRNAs + Mutations	2 + 36 + 28 + 34 *	2	0.74	0.71, 0.77
**UI model M4-B**	**mRNAs + miRNAs + CNVs**	**7 + 38 + 28 + 65**	**40**	**0.91**	**0.88, 0.93**
TCGA model M4-B	mRNAs + miRNAs + CNVs	2 + 36 + 28 + 65 *	2	0.76	0.73, 0.79
**UI model M4-C**	**mRNAs + Mutations + CNVs**	**7 + 38 + 35 + 65**	**42**	**0.91**	**0.88, 0.95**
TCGA model M4-C	mRNAs + Mutations + CNVs	2 + 36 + 34 + 65 *	10	0.75	0.73, 0.78
**UI model M4-D**	**miRNAs + Mutations + CNVs**	**7 + 28 + 35 + 65**	**53**	**0.88**	**0.84, 0.92**
TCGA model M4-D	miRNAs + Mutations + CNVs	2 + 28 + 34 + 65 *	9	0.77	0.74, 0.80
**Replication of Prediction Models using Five Data Classes**					
**Model Number**	**Data classes included** **Clinical +**	**# Input variables**	**# Resulting variables**	**AUC**	**95% CI**
**UI model M5-A**	**mRNAs + miRNAs + Mutations + CNVs**	**7 + 38 + 28 + 35 + 65**	**47**	**0.8**	**0.85, 0.91**
TCGA model M5-A	mRNAs + miRNAs + Mutations + CNVs	2 + 36 + 28 + 34 + 65 *	8	0.76	0.73, 0.78

**Table 4 ijms-20-01205-t004:** Validation of prediction models using data from TCGA EEC dataset. As described in the Methods section, the threshold cut-off values were selected to attain a sensitivity of around 90% and the specificity and negative predictive value. The goal was to create models that would capture at least 90% of the high-risk cases, while ruling out most low risk ones. Recurrence probability scale *: 1/(exp(-score) + 1), where score is the resulting value of the prediction model on a log scale.

	Model M2-B Clinical + miRNAs		Model M2-C Clinical + Mutations		Model M3-C Clinical + mRNAs + Mutations		Model M3-D Clinical + miRNAs + Mutations	
Recurrence probability scale *	Cut-off = 0.5004		Cut-off = 0.4984		Cut-off = 0.7309		Cut-off = 0.5151	
	Value	95% CI	Value	95% CI	Value	95% CI	Value	95% CI
Sensitivity	90%	85%, 94%	90%	86%, 94%	90%	82%, 98%	90%	86%, 94%
Specificity	38%	31%, 44%	16%	8%, 26%	10%	1%, 23%	30%	23%, 37%
Positive Predictive Value (PPV)	56%	51%, 61%	49%	47%, 52%	13%	12%, 15%	53%	50%, 57%
Negative Predictive Value (NPV)	79%	70%, 84%	64%	47%, 74%	87%	45%, 94%	76%	66%, 81%
Accuracy	62%	54%, 68%	51%	47%, 56%	20%	13%, 32%	58%	52%, 63%

**Table 5 ijms-20-01205-t005:** Variables included in the best performing prediction models. Performance of these models was measured by AUC, sensitivity, specificity, and positive and negative prediction values in both the UI (testing) and TCGA (validation) datasets. * Weights for clinical variables were calculated as the exponential of the estimate in the Lasso regression model. Then, the score of the variable is calculated multiplying the weight of the variable by its value. For example, for age, the score is the weight of age, 1.03 times the age in years; for grade, the score is the weight, 12.99, 1.48, or 1.71 times the numerical grade of the tumor (weights differs among the various models). ** Details of individual weights for miRNA expression are in Table A1. # Details of individual weights for somatic mutations are in Table A2. ## Details of individual weights for gene expression are in Table A3.

Prediction Model	M2-B	M2-C	M3-C	M3-D
*Clinical variables*	Weight of clinical variables *
Age	1.03	-	-	-
History of other cancers	0.93	-	-	-
Grade	12.99	1.27	1.01	1.48
BMI	-	0.99	-	-
*Molecular variables*	Log2 transformed and normalized miRNA expression **:	
miRNAs	MIR125B1, MIR181A1, MIR181A2HG, MIR188, MIR301B, MIR30B, MIR3142, MIR345, MIR3690, MIR4269, MIR4307, MIR4463, MIR492, MIR5692A1, MIR578, MIR601, MIR633, MIR6503, MIR6769A, MIR6820			MIR125B1, MIR181A1, MIR181A2HG, MIR188, MIR30B, MIR3690, MIR4269, MIR4307, MIR633, MIR876
Somatic mutations	Number of mutations per gene and person ^#^:	
	*AARS2, ABCD1, ADAMTS13, ATL1, C14orf37, CEP350, CGNL1, COL9A3, CR2, CTAGE8, DAGLA, ENTPD1, FAM111A, HIP1R, HSD17B8, KIF20B, KIZ, LCORL, MAP3K12, MAPKBP1, MPHOSPH8, NOTCH4, NR2C2, PANK2, PCSK5, PIGN, PVR, RPAP1, RSF1, SHROOM2, VDR, ZDHHC24, ZNF780B*	*ADAMTS13, C14orf37, CEP350, CTAGE8, HIP1R, MAPKBP1, NR2C2, PIGN, RSF1, SHROOM2, VDR, ZNF780B*	*AARS2, ABCD1, ADAMTS13, ATL1, C14orf37, CGNL1, COL9A3, CTAGE8, DAGLA, FAM111A, HIP1R, KIZ, LCORL, MAP3K12, MPHOSPH8, NOTCH4, NR2C2, PCSK5, PIGN, PVR, RSF1, SHROOM2, TMEM41B, VDR, ZNF780B*
Gene expression	Log2 transformed and normalized gene expression ^##^:	
		*AQP2, C1QL4, C5orf17, CDH19, COLCA2, FAIM2, FGF18, HAS3, IGFL2, IGFL4, IL23R, LINC01128, LOC101927701, LOC101929529, LONP2, MAN2A2, MRPS28, P4HA2, SCARNA4, SLC25A21, SPATA4, TAC1, TBATA, TFAP2A-AS1, TGFA.IT1, TUBAL3, VAX2, ZNF398*

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
