# Peer review of "A Prediction Model for Preoperative Risk Assessment in Endometrial Cancer Utilizing Clinical and Molecular Variables"

_ijms, 2019, doi:10.3390/ijms20051205_

Round 1
Reviewer 1 Report
Author integrated multi-layer of genomic data to construct the prediction model for EEC patients, which can help to stratify the pre-operative risk of EEC patients. The prediction model including clinical feature, miRNA data and CNV was based on the cross-validation scheme and lasso method which efficiently prevented the overfitting problem and was demonstrated to improve the prediction accuracy comparing to the based prediction model only including the based clinical features.
The main contribution of this study is to utilize the molecular data and clinical data to guide the pre-operative management, and risk evaluation.
Major comments:
1) Have authors considered to use other learning methods instead of lasso, such as elastic net, grouped ssLasso-GLM method and some deep learning method? Perhaps these methods were more helpful to improve the prediction accuracy and looked more reasonable than Lasso, due to the high collinearity existing in genomic data. The AUC of 1 for M2-C model was doubted, maybe it was due to the high false positive. The authors should provide a performance comparison against these methods and models.
2) In the Results part, authors described that M2-B, M2-C, and M3-D performed best, while M3-C (AUC: 95%) seemed to be better than M3-C model (AUC: 94%) according to Table 2 and Table 3. Can authors re-assess these models or carefully check the tables to correct this?
3) In the abstract, authors said DNA methylation was also used to incorporate into the prediction model. However, we can’t see any models or description of the performance of DNA methylation. I was curious whether authors applied the DNA methylation to prediction models, and how about these models’ performance for prediction of EEC risk if it did be applied.
4) Besides the multi-layer genomic data authors used in this study, there were other types of omics-data, such as chip-seq datasets and DNase-seq data. Can authors collect these datasets to incorporate and assess the prediction model?
5) Can authors provide the full scripts to construct and perform the prediction procedure?
Minor comments:
a) The authors mark the non-significant p values in Table 5, while it might be not appropriately from the statistical angle. Can the authors change to mark the significant p values?
b) Figure 3 was missing.
c) Figure 4 legend was missing.
d) In the 4.2.2, I thought “Supplementary Table B2” pointed to the “Supplementary Table S2” but the captions of tables were different. Authors should carefully check the full manuscript and make the figures and tables consistent in the full manuscript.
Author Response
Salinas, et al., “A prediction model for preoperative risk assessment in endometrial cancer utilizing clinical and molecular variables”
Response to Reviewers
We appreciate the reviewers’ careful reading and constructive critiques. We appreciate the suggestions to expand our descriptions of methods and rationale for methodological choices in order to improve the clarity and readability of the manuscript. We believe we have fully addressed all critiques in this revised manuscript. Point-by-point responses are provided below.
Reviewer 1
Comments and Suggestions for Authors
Author integrated multi-layer of genomic data to construct the prediction model for EEC patients, which can help to stratify the pre-operative risk of EEC patients. The prediction model including clinical feature, miRNA data and CNV was based on the cross-validation scheme and lasso method which efficiently prevented the overfitting problem and was demonstrated to improve the prediction accuracy comparing to the based prediction model only including the based clinical features.
The main contribution of this study is to utilize the molecular data and clinical data to guide the pre-operative management, and risk evaluation.
Major comments:
1) Have authors considered to use other learning methods instead of lasso, such as elastic net, grouped ssLasso-GLM method and some deep learning method? Perhaps these methods were more helpful to improve the prediction accuracy and looked more reasonable than Lasso, due to the high collinearity existing in genomic data. The AUC of 1 for M2-C model was doubted, maybe it was due to the high false positive. The authors should provide a performance comparison against these methods and models.
We agree with the importance of using different machine learning methods, which we have explored. In previous studies we constructed prediction models with up to 9 different statistical methods, including those proposed by the reviewer (see Gonzalez Bosquet, J.; Newtson, A. M.; Chung, R. K.; Thiel, K. W.; Ginader, T.; Goodheart, M. J.; Leslie, K. K.; Smith, B. J., Prediction of chemo-response in serous ovarian cancer. Molecular cancer 2016, 15, (1), 66.). Based on the results of that study, it was our conclusion that Lasso was our preferred method because consistently handles missing values and lower numbers of samples. Lasso also computes the AUC without reporting any errors. Also, other groups like TCGA network that integrated molecular information also used Lasso to reduce dimensionality in their cluster analyses for ovarian and endometrial cancer. The iCluster and iClusterPlus clustering analyses use Lasso regression analysis for their variable selection (reference 26).
We now reference previous prediction models in ovarian cancer using different methods to explain the selection of Lasso as the preferred machine learning method and (see Section 4.4.3. Prediction Model Construction).
2) In the Results part, authors described that M2-B, M2-C, and M3-D performed best, while M3-C (AUC: 95%) seemed to be better than M3-C model (AUC: 94%) according to Table 2 and Table 3. Can authors re-assess these models or carefully check the tables to correct this?
We have re-assessed the best models for UI, TCGA replication and TCGA validation to be consistent with their performances based on AUCs (). The decision to select the best potential prediction model for further validation was based on the following parameters:
- First: performance of the model, based on AUC, on the UI dataset;
- Second: performance of the model, based on AUC, on the replication set in TCGA;
The decision to exclude models with gene expression was subjective. We now include the selected models for validation model M3-C, as suggested by the reviewer. Also, we explain better the model selection process. Please note thatthe tables are now Tables 4 and 5 in the revised manuscript. Also, Supplementary Tables A1, A2, and A3 have been modified accordingly.
3) In the abstract, authors said DNA methylation was also used to incorporate into the prediction model. However, we can’t see any models or description of the performance of DNA methylation. I was curious whether authors applied the DNA methylation to prediction models, and how about these models’ performance for prediction of EEC risk if it did be applied.
We apologize for this error. DNA methylation data were not included in the prediction model. This text has been deleted. We agree that adding DNA methylation data will provide an interesting dimension to the prediction models; we are in the process of seeking funding to obtain these data.
4) Besides the multi-layer genomic data authors used in this study, there were other types of omics-data, such as chip-seq datasets and DNase-seq data. Can authors collect these datasets to incorporate and assess the prediction model?
As with DNA methylation data, we will love to add these datasets to our prediction models. We are asking for more funding to proceed with more molecular analyses since Chip-seq and DNA-seq cannot be inferred from RNA-seq.
5) Can authors provide the full scripts to construct and perform the prediction procedure?
As requested, we have added an Appendix C with a sample of the code so the analysis can be replicated. [Jesus – you could also provide your programs on GitHub]
Minor comments:
a) The authors mark the non-significant p values in Table 5, while it might be not appropriately from the statistical angle. Can the authors change to mark the significant p values?
This has been corrected.
b) Figure 3 was missing.
(Now Figure 4) There was a problem when converting from Word document to PDF during the editing process. The editing team was notified, and it has been solved.
c) Figure 4 legend was missing.
(Now Figure 5) Same as above.
d) In the 4.2.2, I thought “Supplementary Table B2” pointed to the “Supplementary Table S2” but the captions of tables were different. Authors should carefully check the full manuscript and make the figures and tables consistent in the full manuscript.
We apologize for this inadvertent mistake. Figure and table numbers have been corrected throughout the manuscript.
Reviewer 2 Report
In the manuscript "A prediction model for preoperative risk assessment in endometrial cancer utilizing clinical and molecular variables" by Salinas et al., the authors describe an approach to improve preoperative risk assessment in endometrial cancer patients by combining clinico-pahological data as well as certain molecular variables determined by RNA-seq analyses. This approach was performed in a test cohort and in TCGA/GEO validation cohorts. Prediction models that included clinical data, miRNA expression and/or somatic mutation data outperformed single factor models. Overall, the presented data are interesting, however, some points (especially writing of the Results section) need to be improved.
Comments:
1. Line 28: The word "data" is missing after "somatic mutation".
2. The survival plot for TCGA low/high risk patients is missing.
3. Line 99: "A total of 62 primary EEC patient tumor tissues produced RNA of sufficient..." should be something like "RNA extracted from primary tumor tissue samples of 62 EEC patients was of suffiecient quality for RNA-seq analyses..."
4. How was RNA-seq quality checked?
5. What does cross-validation selection mean? This step and all results from RNA-seq analyses (differential mRNA expression, differential miRNA expression, identification/selection of 398 somatic mutations) need to be descibed in more detail.
6. Figure 2: Which cut-offs were used for selection? What does the colour code mean? Legend to this Figure needs to be more detailed!
7. What is the difference between replication set and validation set?
8. Why was the GEO dataset included? It contains only clinical and mRNA expression data, no miRNA expression, mutation or copy number alteration data are included in this dataset.
9. Table 1.: It is unclear for the reader how "resulting variables" were selected. This process needs to be clearly descrbed in the results section.
10. Line 133: "databases" should be "datasets"
11. Accodring to the cBio data portal for TCGA data, FAM134B = RETREG1 and LOC101927701 = LINC01812. Thus, data about these genes are available and analyses should be re-calculated.
12. Table 4: "Performing" should be "Performance"
13. Line 215: there is something missing in "involvement, o"
14. Legend to Figure 4 is missing. In addition, I suggest to use Figure 4 as the first Figure.
15. I suggest to use Table 5 as Table 1.
16. Lines 307-309: "We have determined...." This is nice information but does not fit into the Methods section of a manuscript.
17. Line 321: "such as TopHat2" sounds like the authors do not know which algorithm was indeen used for alignment.
18. Why was the cut-off in the Cox regression set to 0.1 and not 0.05?
Author Response
Reviewer 2
Comments and Suggestions for Authors
In the manuscript "A prediction model for preoperative risk assessment in endometrial cancer utilizing clinical and molecular variables" by Salinas et al., the authors describe an approach to improve preoperative risk assessment in endometrial cancer patients by combining clinico-pathological data as well as certain molecular variables determined by RNA-seq analyses. This approach was performed in a test cohort and in TCGA/GEO validation cohorts. Prediction models that included clinical data, miRNA expression and/or somatic mutation data outperformed single factor models. Overall, the presented data are interesting, however, some points (especially writing of the Results section) need to be improved.
Comments:
1. Line 28: The word "data" is missing after "somatic mutation".
This mistake has been corrected.
2. The survival plot for TCGA low/high risk patients is missing.
TCGA survival curves have been added to the revised manuscript as a new Figure 2.
3. Line 99: "A total of 62 primary EEC patient tumor tissues produced RNA of sufficient..." should be something like "RNA extracted from primary tumor tissue samples of 62 EEC patients was of suffiecient quality for RNA-seq analyses..."
We have edited this sentence per the reviewer’s suggestion.
4. How was RNA-seq quality checked?
This information is provided in the second paragraph of the section ‘4.3.1. University of Iowa (UI)’. Briefly, we perform quality control as previously reported in the following reference: Schroeder, A.; Mueller, O.; Stocker, S.; Salowsky, R.; Leiber, M.; Gassmann, M.; Lightfoot, S.; Menzel, W.; Granzow, M.; Ragg, T., The RIN: an RNA integrity number for assigning integrity values to RNA measurements. BMC Mol Biol 2006, 7, 3.
5. What does cross-validation selection mean? This step and all results from RNA-seq analyses (differential mRNA expression, differential miRNA expression, identification/selection of 398 somatic mutations) need to be described in more detail.
We apologize that we did not provide sufficient explanation of our validation methods. In the revised manuscript, we now clarify the variable selection process. We have added the following explanation to ‘4.4.2. Variable selection for prediction modelling’ section:
Before the prediction model was constructed, we selected those variables to be introduced in the model with simple univariate statistical methods. We only use those features that passed the univariate statistical criterion in the subsequent model steps. The caret package fits a simple linear model between a single feature and the outcome, then the p-value for the whole model F-test is returned. Predictors that have statistically significant differences (with a p-value < 0.05) between the classes (low and high-risk) were then used for modeling. However, cross-validation of the subsequent models is biased unless some sort of resampling is included in the feature selection step. To address this for classification models, the package caret includes a cross-validation feature that achieves this resampling. Including cross-validation on the selection of variables to include in the model and in the construction of the prediction model decreases the possibility of overfitting the final model.
We have also added a new reference to explain overfitting in the prediction process: Subramanian, J.; Simon, R., Overfitting in prediction models - is it a problem only in high dimensions? Contemp Clin Trials 2013, 36, (2), 636-41.
6. Figure 2: Which cut-offs were used for selection? What does the colour code mean? Legend to this Figure needs to be more detailed!
We appreciate this suggestion and have improved the legend by incorporating information about the method of variable selection and cut-offs. We also provide an explanation about color coding and the range of the data they represent. Note that these data are now Figure 3 in the revised manuscript
7. What is the difference between replication set and validation set?
We have expanded section 4.4.4 ‘The Cancer Genome Atlas Replication and Validation’ to better explain the differences between replication and validation in section 4.4.4.. Briefly, ‘replication’ denotes repeating the prediction analysis that performed in the UIHC dataset in TCGA dataset, using the same variables and lasso method. For ‘validation’ of the UIHC dataset in TCGA dataset, we take the model that was built in UI and plug in the TCGA data to see if it can discriminate between low and high-risk patients in TCGA. The validation does an inference: with UI-built model and TCGA data, the model tries to predict TCGA classes.
8. Why was the GEO dataset included? It contains only clinical and mRNA expression data, no miRNA expression, mutation or copy number alteration data are included in this dataset.
A priori we did not know which molecular variables were going to improve the clinical prediction model. After all prediction models were built, we opted to keep GEO models for two reasons:
- It corroborated the good performance of a model with only clinical variables in another independent dataset;
- It agreed with UIHC and TCGA prediction models that adding only gene expression to clinical data did not improve the prediction model significantly.
9. Table 1.: It is unclear for the reader how "resulting variables" were selected. This process needs to be clearly descrbed in the results section.
We have now expanded the legend for this table (now Table 2 in the revised manuscript) as well as Results section to better explain how resulting variables were selected. The lasso method discards all variables that do not contribute to the prediction model. For example, including one type of data, miRNA expression: we introduced all 55 variables (Input variables) in the lasso model, but only 28 of them (Resulting variables) were informative in a prediction model (AUC of 84%). These Resulting variables, i.e., the most informative of all miRNAs, were subsequently used for all combinations where miRNA data were incorporated. The same was done for the other molecular variables, resulting 38 mRNAs, 35 somatic mutations and 65 CNVs, and also clinical data (7 variables).
In models with 2 or more types of data, the Input variables were the more informative variables (i.e., Resulting variables) from the prediction process in the individual prediction analyses.
10. Line 133: "databases" should be "datasets"
This mistake has been corrected.
11. According to the cBio data portal for TCGA data, FAM134B = RETREG1 and LOC101927701 = LINC01812. Thus, data about these genes are available and analyses should be re-calculated.
These genes were not present in the database of 25,673 genes from TCGA endometrioid endometrial cancer when data were processed as follows: FASTQ files from TCGA data were aligned with the hg38 version of the human genomes. Next, BAM files were created and downloaded. Gene expression was extracted with featureCount. In addition, even before normalization and filtering for missing values (genes with more than 50% of missing values were excluded from the analysis), there were no expression values for FAM134B (or RETREG1, or JK1, or JK-1, other valid genes from this transcript) or LOC101927701 (or LINC01812, another valid name). To address this reviewer’s critique, we have edited the manuscript to include the following sentence:
Part of this decline in performance may be because two selected transcripts in the UI model (FAM134B and LOC101927701, a non-coding RNA) had no expression data in TCGA dataset.
12. Table 4: "Performing" should be "Performance"
Corrected
13. Line 215: there is something missing in "involvement, o"
It was part of the high-risk classification inclusion criteria. We have amended the text as follows:
We included as high-risk patients those with adnexal or cervical involvement, or patients with other local and/or distant metastasis.
14. Legend to Figure 4 is missing. In addition, I suggest to use Figure 4 as the first Figure.
There was a problem when converting from Word document to PDF during the editing process. The editing team was notified, and it has been resolved. Per this reviewer’s recommendation, Figure 4 is now Figure 1 and all other figures have been renumbered accordingly.
15. I suggest to use Table 5 as Table 1.
We have made this adjustment as suggested.
16. Lines 307-309: "We have determined...." This is nice information but does not fit into the Methods section of a manuscript.
We deleted the referred sentence from the Methods section.
17. Line 321: "such as TopHat2" sounds like the authors do not know which algorithm was indeen used for alignment.
We gave an example that we thought the reader would be more familiar with. However, the reviewer is right, and there have been noticeable improvements in software for genetic alignment. For this study we used STAR, a new alignment algorithm that is faster, more efficient and more accurate than TopHat version 2.0.
We changed the alignment source and added the corresponding reference (Dobin, A.; Davis, C. A.; Schlesinger, F.; Drenkow, J.; Zaleski, C.; Jha, S.; Batut, P.; Chaisson, M.; Gingeras, T. R., STAR: ultrafast universal RNA-seq aligner. Bioinformatics 2013, 29, (1), 15-21).
18. Why was the cut-off in the Cox regression set to 0.1 and not 0.05?
This was an error. We used the p-value cut-off of 0.05, as the reviewer noted. In fact, in the multivariate analysis we only included those variables with a p-value < 0.05 (now Figure 2B).
Round 2
Reviewer 1 Report
The manuscript has been improved. However, the authors didn't fully address the concerns. there were still some concerns about the method and results.
1. All tables in the results were not standard format and not suit for publication.
2. Authors used caret to conduct variable selection first. Lasso can conduct variable selection as well. Why did authors use caret first? It was not clear demonstration in Methods.
3. I still doubt the results in Table 2 and 3. The AUC of the model “Mutation” can reach 1, which was worthy to doubt. Can authors use other methods to compare or verify the AUC results?
Author Response
Reviewer 1
The manuscript has been improved. However, the authors didn't fully address the concerns. there were still some concerns about the method and results
1. All tables in the results were not standard format and not suit for publication.
The Editorial Team at IJMS has modified the tables to align with the journal’s preferences.
2. Authors used caret to conduct variable selection first. Lasso can conduct variable selection as well. Why did authors use caret first? It was not clear demonstration in Methods.
Caret is a software package that performs univariate selection of prediction variables with cross-validation. Lasso is a multivariate regression model for prediction.
There were two approaches that we could have followed, both of which would be valid: 1) a lasso prediction model with all variables; 2) a univariate analysis first to decrease the number of variables, followed by the lasso prediction model.
The first approach tends to include more variables in the resulting prediction model, including some that may not be predictive, leading to overfitting. We opted for the second method because it tends to result in a prediction model with less variables (i.e., a more sparse model), that would be easier to validate retrospectively and prospectively. Thus, we applied the caret package to select variables that were informative for the prediction model of risk (at a univariate p-value < 0.05) with a k-fold cross-validation. The significant variables in caret analyses were then introduced in a multivariate lasso regression model of prediction.
We clarified these concepts in the Material and Methods section (4.4.2. Variable selection for prediction modeling).
3. I still doubt the results in Table 2 and 3. The AUC of the model “Mutation” can reach 1, which was worthy to doubt. Can authors use other methods to compare or verify the AUC results?
In a previous published study we constructed prediction models with up to 9 different statistical methods, including those proposed by the reviewer (see: Gonzalez Bosquet, J.; Newtson, A. M.; Chung, R. K.; Thiel, K. W.; Ginader, T.; Goodheart, M. J.; Leslie, K. K.; Smith, B. J., Prediction of chemo-response in serous ovarian cancer. Molecular cancer 2016, 15, (1), 66.). Based on the results of that study, it was our conclusion that lasso was our preferred method because had similar result to the other methods, was able to handle missing values, performed variables selection with simpler resulting models, was able to integrate simultaneously binomial and continuous variables, and determine areas under the curve consistently under diverse conditions.
Instead of validating our prediction models from the UI (training set) with other statistical methods, we replicated and validated our results in an independent dataset, TCGA (testing set). The performance of all models in the training set was superior to the testing set. There are potentially numerous reasons why there could be differences between prediction performances of the training and testing sets (e.g., see Reference 42). In the third paragraph of the Discussion section we commented some of these possibilities, but we should have mentioned also about overfitting. Overfitting, as we see in our results, is characterized by a high accuracy for a classifier when evaluated on the training set but a low accuracy when evaluated on the testing set. Overfitting has been recognized as a problem when there are more variables than samples (Simon R, Radmacher MD, Dobbin K, McShane LM. Pitfalls in the use of DNA microarray data for diagnostic and prognostic classification. J Natl Cancer Inst. 2003;95:14-18). This may happen even when the initial prediction model is performed with cross-validation, with sampling and re-sampling.
We have now added these comments about overfitting in the third paragraph of the Discussion section, and add the above reference about overfitting.
Reviewer 2 Report
The authors addressed my concerns.
Author Response
Reviewer 2:
The authors addressed my concerns.
Round 3
Reviewer 1 Report
Authors fully addressed my concerns. The only thing that authors need to carefully check the writing in the manuscript such as lasso should be Lasso.